# Attenuation of Sepsis-Induced Acute Kidney Injury by Exogenous H_2_S via Inhibition of Ferroptosis

**DOI:** 10.3390/molecules28124770

**Published:** 2023-06-14

**Authors:** Li Zhang, Jin Rao, Xuwen Liu, Xuefu Wang, Changnan Wang, Shangxi Fu, Jian Xiao

**Affiliations:** 1School of Medicine, Guangxi University, Nanning 530004, China; 18957427692@163.com (L.Z.); 18851827679@163.com (X.L.); 2Department of Cardiothoracic Surgery, Changzheng Hospital, Naval Medical University, Shanghai 200003, China; csraojin@163.com; 3School of Health Sciences and Engineering, University of Shanghai for Science and Technology, Shanghai 200093, China; 213332652@st.usst.edu.cn; 4School of Life Sciences, Shanghai University, Shanghai 200444, China; wangchangnan@shu.edu.cn; 5Department of Urology, Kidney Transplantation Center, Ruijin Hospital, Shanghai Jiaotong University School of Medicine, Shanghai 200025, China

**Keywords:** ferroptosis, sepsis, GYY4137, exogenous H_2_S, acute kidney injury, cecal ligation and puncture, mitochondrial oxidative stress

## Abstract

Sepsis-associated acute kidney injury (SA-AKI) results in significant morbidity and mortality, and ferroptosis may play a role in its pathogenesis. Our aim was to examine the effect of exogenous H_2_S (GYY4137) on ferroptosis and AKI in in vivo and in vitro models of sepsis and explore the possible mechanism involved. Sepsis was induced by cecal ligation and puncture (CLP) in male C57BL/6 mice, which were randomly divided into the sham, CLP, and CLP + GYY4137 group. The indicators of SA-AKI were most prominent at 24 h after CLP, and analysis of the protein expression of ferroptosis indicators showed that ferroptosis was also exacerbated at 24 h after CLP. Moreover, the level of the endogenous H_2_S synthase CSE (Cystathionine-γ-lyase) and endogenous H_2_S significantly decreased after CLP. Treatment with GYY4137 reversed or attenuated all these changes. In the in vitro experiments, LPS was used to simulate SA-AKI in mouse renal glomerular endothelial cells (MRGECs). Measurement of ferroptosis-related markers and products of mitochondrial oxidative stress showed that GYY4137 could attenuate ferroptosis and regulate mitochondrial oxidative stress. These findings imply that GYY4137 alleviates SA-AKI by inhibiting ferroptosis triggered by excessive mitochondrial oxidative stress. Thus, GYY4137 may be an effective drug for the clinical treatment of SA-AKI.

## 1. Introduction

Sepsis is a serious organ dysfunction caused by infection, and it is one of the most common diagnoses in intensive care units and the main cause of death in patients with severe illness [1,2]. Bacterial endotoxins and inflammatory reactions in patients with sepsis result in the production of a large number of reactive oxygen species (ROS), which lead to damage and dysfunction of multiple cell types [3]. The kidney is the most vulnerable organ in sepsis, and clinical and basic research has shown that the kidney is dramatically affected during sepsis [4,5]. Acute kidney injury (AKI) is defined as an abrupt decrease in renal function (within 48 h), and it accounts for 15% of intensive care unit admissions worldwide [6,7]. AKI is primarily caused by ischemia and reperfusion injury (IRI), which temporarily blocks blood flow, increases inflammatory processes and induces oxidative stress [8]. The mortality of patients with sepsis increases significantly once renal injury occurs, and mortality of sepsis-associated acute kidney injury (SA-AKI) may reach up to 70% [9]. Therefore, exploring the mechanism of sepsis-related renal injury and reducing renal cell death has become one of the hot topics in the field of kidney therapy. 

Hydrogen sulfide (H_2_S), as a gaseous signaling molecule, is widely involved in physiological metabolic reactions in the body, and also plays an important regulatory role in cooperation with nitric oxide and carbon monoxide [10]. Previous studies have shown that H_2_S plays an important regulatory role in the cardiovascular system, the nervous system, and the protection of the circulatory system from oxidative damage [11]. In recent years, it has been found that AKI results in the downregulation of endogenous H_2_S production, reduces glutathione (GSH) levels, and increases oxidative stress in the heart [12]. H_2_S has been extensively used in various animal and cell models of AKI and proven to ameliorate kidney damage [13]. For example, endogenous H_2_S can ameliorate lipopolysaccharide (LPS)-induced AKI by inhibiting inflammation and oxidative stress via the TLR4/NLRP3 signaling pathway [14], and exogenous H_2_S (in the form of NaHS) can also provide protection from LPS-induced AKI by promoting autophagy and inhibiting apoptosis and the release of inflammatory factors [15]. In addition, it has been reported that H_2_S inhibits IRI-induced AKI by suppressing the NLRP3 inflammasome/Caspase-1 axis [16]; furthermore, H_2_S attenuates CISP-induced AKI by preventing mitochondrial dysfunction via SIRT3 sulfhydrylation [17]. It has also been reported that exogenous H_2_S ameliorates IRI-induced AKI in the elderly by increasing H_2_S levels, reducing macrophage-mediated injury, and promoting the repair process through the inhibition of miR-21 [18]. Although numerous studies have demonstrated the protective effects of H_2_S during diverse pathological processes associated with AKI [19], there is little information about the specific mechanism by which H_2_S alleviates AKI and it remains to be further studied. 

Ferroptosis is a novel form of programmed cell death that was proposed in 2012 [20]. Unlike other known cell death modalities (apoptosis, necrosis, pyroptosis, and autophagy), ferroptosis is iron-dependent and the morphology of cells is characterized by mitochondrial clustering, increased mitochondrial membrane density, reduction/absence of mitochondrial cristae, and no nuclear condensation or chromatin margination in cancer cells [21,22]. Excessive accumulation of intracellular lipid peroxides and free iron ions can lead to ferroptosis [23,24]. Ferroptosis is regulated by multiple cellular metabolic pathways, including redox homeostasis; iron metabolism; mitochondrial activity; and amino acid, lipid, and sugar metabolism [25]. Glutathione peroxidase 4 (Gpx4) acts as an essential regulator of ferroptosis by inhibiting lipid peroxidation [26]. Transferrin (TF) is the main source of iron in tissues, involved in the transport of extracellular iron into the cell [27]. Ferritin heavy polypeptide1 (FTH-1) plays an important role in maintaining the iron balance in cytoplasm and regulating ferroptosis [28], and it plays a protective role in sepsis-induced organ failure [29]. In the last decade, the research on ferroptosis has been expanding exponentially, and many studies have revealed that ferroptosis is widely involved in abnormal metabolic and biochemical processes in vivo and plays an important role in pathological processes such as cancer, inflammation, neurodegeneration, renal failure, and cardiac IRI [30]. Recent studies have also shown that ferroptosis and AKI are closely related. For example, the relationship between AKI and iron metabolism was described in a study by Scindia et al., who reported that AKI induced by IRI caused ferroptosis in kidney cells [31]. With regard to the relationship between H_2_S and ferroptosis, current studies indicate that downregulation of the H_2_S synthase CSE and H_2_S signaling can contribute to mitochondrial damage, abnormal lipid metabolism, membrane lipid peroxidation, and ferroptotic cell death; in addition, exogenously applied H_2_S can block RSL3-induced ferroptotic myoblast death by inhibiting ALOX12 acetylation [32]. H_2_S was also demonstrated to attenuate sepsis-induced myocardial cell and tissue injury by significantly regulating the BECN1 signaling pathway and significantly decreasing myocardial ferroptosis [33]. Moreover, H_2_S-regulated iron metabolism reduced oxidative stress levels in cardiomyocytes, inhibited cardiomyocyte ferroptosis, and protected cardiac function in aging rats [34]. With regard to the underlying mechanism, H_2_S protected retinal pigment epithelium cells against ferroptosis through the AMPK- and p62-dependent non-canonical NRF2–KEAP1 pathway [35]. However, the exact biomolecular mechanisms are unclear, and the involvement of ferroptosis in H_2_S-induced alleviation of AKI has not been reported so far.

On the basis of the findings reported so far, in the present study, we evaluated the potential therapeutic utility of the exogenous H_2_S GYY4137, which can release H_2_S in a stable manner [36], for the treatment of SA-AKI and explored the involvement of ferroptosis in the underlying mechanism. These findings will make an important contribution to this field, as the role of ferroptosis in H_2_S treatment of SA-AKI has not been explored so far. 

## 2. Result

### 2.1. Increase in Ferroptosis in CLP-Induced SA-AKI Model Mice

To verify that AKI was caused by CLP-induced sepsis in the model mice, we measured and compared the level of Cre and BUN in the sham and CLP treatment groups at 12, 24, 48, and 72 h, as they are conventionally considered as critical markers of renal function. At all the measured time points, the serum Cre and BUN levels were significantly higher in the model mice than in the sham mice, with the peak BUN and Cre levels detected at 24 h in the model mice (Figure 1A,B). Therefore, we obtained samples from the sham and CLP-treated groups at 24 h after the CLP procedure for H&E staining (Figure 1C,D). No pathological changes were observed in the kidney tissue of the sham groups (Figure 1C), but significant inflammatory infiltration and massive glomerular cell detachment were noted in the CLP-treated groups (Figure 1D). In addition, we observed smaller mitochondria, absence of cristae or a decrease in their size, and broken outer membranes in the CLP groups compared with the sham groups under a transmission electron microscope (Figure 1E,F).

To explore the potential role of ferroptosis in SA-AKI, we measured the expression levels of the traditional ferroptosis marker proteins Gpx4, FTH-1, and TF in the sham and CLP-treated groups at 12, 24, 48, and 72 h (Figure 1G). The protein level of Gpx4, which provides protection against membrane lipid peroxidation, was generally lower in the CLP-treated groups than in the corresponding sham group, and this difference was most significant at 24 h after CLP (Figure 1H). In contrast, the protein expression levels of FTH-1 (an iron storage protein found in the cytoplasm) and TF (an iron transporter that prevents the formation of ROS) were generally higher than that in sham mice at 24 h (Figure 1I,J). The data above demonstrate that CLP leads to SA-AKI and aggravation of ferroptosis in CLP model mice with AKI.

### 2.2. Role of Ferroptosis in LPS-Induced AKI under In Vitro Conditions

LPS-treated MRGECs were utilized to simulate SA-AKI under in vitro settings. Because the BUN and Cre levels and the degree of ferroptosis were highest at 24 h in the above experiment, measurement was performed at 24 h in the following experiments. We performed a CCK-8 assay, and the results showed that 100 µg/mL LPS treatment significantly inhibited MRGECs viability at 24 h, and cell viability that was decreased upon treatment was most likely due to ferroptosis (Figure 2A), but after the application of 10 µM ferroptosis inhibitor Fer-1, the MRGECs viability stimulated by 100 µg/mL LPS was significantly increased (Figure 2B). Furthermore, measurement of ferroptosis-related protein expression levels in MRGECs stimulated by LPS at concentrations of 0, 0.1, 1, 10, and 100 µg/mL for 24 h (Figure 2C) showed that the intracellular ferroptosis level increased to different degrees under stimulation with different concentrations of LPS, with the greatest increase observed at an LPS concentration of 100 µg/mL (Figure 2D–F). Treatment with the ferroptosis inhibitor, Fer-1 at 10 µM was found to be effective in reversing 100 µg/mL LPS-induced ferroptosis in normal MRGECs at 24 h, as indicated by an increase in Gpx4 protein levels and a decrease in TF and FTH-1 protein levels compared to the 100 µg/mL LPS-treated group that was not exposed to Fer-1 (Figure 2G–J).

### 2.3. Attenuation of AKI and Ferroptosis in CLP Model Mice Treated with GYY

In CLP-induced sepsis mice, the serum H_2_S levels were significantly lower than those in the sham-operated groups at 24 h after CLP treatment (Figure 3C), but GYY treatment was found to restore the serum H_2_S level in the CLP + GYY4137 group (Figure 3I). CSE is an endogenous H_2_S synthase that plays an important role in H_2_S production in the kidney. Therefore, the expression of this enzyme was measured in the kidney samples. The protein expression of CSE significantly decreased in CLP-induced SA-AKI model mice compared with the sham mice at 24 h after CLP treatment (Figure 3A,B). To determine the optimal therapeutic concentration, we designed a concentration gradient of GYY (5, 10, 25, and 50 mg/kg). The results showed that the CLP-operated mice that received GYY treatment exhibited an increase in the protein expression of Gpx4 and a decrease in the expression of TF and FTH-1 compared with the untreated CLP groups, and this was especially evident when the concentration of GYY was 10 mg/kg (Figure 3D–G). The above data suggest that GYY treatment attenuated sepsis-induced ferroptosis in mice with AKI in a dose-dependent manner. Measurement of the level of Fe^2+^ in the kidney and BUN and Cre in the serum showed that treatment with 10 mg/kg GYY could alleviate the increase in their levels in CLP model mice (Figure 3H,J,K). Moreover, 10 mg/kg GYY treatment after CLP could mitigate inflammatory infiltration and massive glomerular cell detachment in the CLP model mice, as indicated by H&E staining (Figure 3L–N), and reverse the decrease in mitochondrial size, the reduction or disappearance of cristae, and fragmentation of the outer membranes, as observed under a transmission electron microscope (Figure 3O–Q). 

### 2.4. Attenuation of AKI and Ferroptosis by H_2_S under In Vitro Conditions

Consistent with the findings of the in vivo experiments in mice, protein expression of CSE was markedly decreased in the MRGECs after 24 h of 100 µg/mL LPS stimulation compared with the control cells (Figure 4A,B), and this was accompanied by a decline in the level of H_2_S in normal MRGECs (Figure 4C). To determine the optimal therapeutic concentration under in vitro conditions, we designed a concentration gradient for GYY (0, 0.2, 0.4, 0.8, and 1 mM) that was used to treat cells exposed to 100 µg/mL LPS. The protein expression of Gpx4 was significantly decreased on treatment with 0.8 mM GYY, and the protein expression of both TF and FTH-1 was significantly increased at 24 h after LPS stimulation compared with the LPS-treatment group without GYY treatment (Figure 4D–G). As shown in Figure 4, GYY (0.8 mM) treatment promoted the cell viability of MRGECs (Figure 4I) and reversed the decrease in the intracellular level of H_2_S stimulated by 100 µg/mL LPS (Figure 4H). 

### 2.5. Attenuation of Ferroptosis by H_2_S via Regulation of Mitochondrial Oxidative Stress under In Vitro Conditions

Mito-TEMPO is a mitochondrial-targeted superoxide dismutase mimetic with the ability to scavenge superoxide and alkyl radicals. When 100 µg/mL LPS-treated MRGECs (24 h) were treated with Mito-TEMPO at 10 µM, their cell viability was significantly higher than that of LPS-treated cells that were not treated with Mito-TEMPO (Figure 5A). We examined the relative values of GSH, MDA, ROS, and MMP associated with LPS-induced AKI in MRGECs. GSH and MMP were significantly decreased (Figure 5D,E) and the level of mitochondrial ROS and MDA were significantly increased in the CLP groups (Figure 5B,C), as compared with the control group. Mito-TEMPO treatment at 10 µM could alleviate the abnormal changes in GSH, MDA, mitochondrial ROS, and MMP in the LPS-treated group (Figure 5B–E), and these were similar to the effects of 0.8 mM GYY in MRGECs (Figure 5F–I). In addition, Mito-TEMPO could also alleviate ferroptosis triggered by 100 µg/mL LPS in MRGECs, as evidenced by a decrease in Gpx4 and an increase in FTH-1 and TF (Figure 5J–M).

## 3. Discussion

In the present study, we have demonstrated for the first time that GYY4137, an exogenous form of H_2_S, induces an increase in H_2_S levels in septic kidney cells and alleviates AKI by inhibiting ferroptosis induced by mitochondrial oxidative stress.

CLP was used to induce sepsis in model mice because this model can simulate the complexity of sepsis in humans better than other known models [37]. The CLP model is, therefore, widely used in research on the pathophysiology of sepsis and is considered to be the key preclinical trial model for research into any new treatment of human sepsis, as well as the gold standard model for sepsis research [38,39]. In our study, ligation was performed at about one-third of the cecum in the mice, and a small amount of the cecal content was squeezed out through two fine pores with a 4-point needle. Many studies have reported that LPS treatment can be used to establish a septic myocardial cell injury model [40]. Therefore, in our study, the MRGECs was selected and treated with LPS according to the preliminary experimental results. Kidney tissue and cell culture medium supernatant were collected 24 h after various control and experimental treatments for analysis. Our in vivo experiments demonstrated that intraperitoneal administration of GYY after CLP treatment for 30 min significantly improved the survival rate of the model mice in a concentration-dependent manner. Further, GYY also restored BUN and Cre to normal levels in CLP model mice with AKI and alleviated the pathological changes in kidney tissues. Accordingly, in the in vitro experiments, GYY was found to improve cell viability and alleviate mitochondrial oxidative stress. These findings imply that H_2_S plays a critical role in the maintenance of normal renal cell function in sepsis-induced AKI, and they are consistent with previously published studies which have indicated that H_2_S has a therapeutic effect on AKI.

It has been reported that H_2_S can inhibit cellular ferroptosis by reducing lipid peroxidation caused by mitochondrial oxidative stress. In a study by Wang et al., the ferroptosis agonist, RSL3 inhibited the expression of Gpx4 and blocked CSE/H_2_S signaling, leading to the intensification of oxidative stress, lipid peroxidation, and cellular ferroptosis, but the ferroptosis antagonist Fer-1 and exogenous NaHS upregulated CSE expression, repaired mitochondrial damage, scavenged ROS production and lipid peroxidation, and improved cell viability, preventing RSL3-induced cellular ferroptosis [32]. Further, exogenous H_2_S has been found to modulate intracellular redox reactions and ameliorate retinal degenerative diseases by inhibiting oxidative stress-induced ferroptosis in retinal pigment epithelium cells [35]. The results of our study show that ferroptosis was exacerbated in animal and cellular models of sepsis, as demonstrated by the decrease in the protein expression of Gpx4, increase in the protein expression of TF and FTH-1, and increase in serum Fe^2+^ levels compared with the sham groups. Furthermore, in the in vitro sepsis model, these effects were reversed by treatment with the ferroptosis inhibitor, Fer-1 and Mito-TEMPO (a mitochondrial superoxide scavenger) [41]. In addition, we found that ferroptosis affected the indicators of mitochondrial oxidative stress, including ROS and MDA, as well as GSH and MMP, in the in vivo and in vitro sepsis models, and these changes were reversed by Mito-TEMPO in the in vitro sepsis model. Interestingly, our results showed that treatment with GYY in the animal and cellular sepsis models could effectively reverse the changes induced in markers of ferroptosis and mitochondrial oxidative stress by sepsis-induced AKI. These results suggest that sepsis may induce ferroptosis by increasing mitochondrial lipid peroxidation and MDA levels and decreasing MMP and GSH, and H_2_S may alleviate ferroptosis-induced AKI by inhibiting mitochondrial oxidative stress.

One of the limitations of this study is that it does not provide information about the molecular mechanisms underlying the protective effect of H_2_S in sepsis. As a result, further experiments are needed regarding the molecular mechanisms underlying the mitigation of SA-AKI by H_2_S via regulation of ferroptosis. Despite this, the findings do provide some level of evidence for the potential benefits of treatment with exogenous H_2_S in sepsis-related AKI. 

To conclude, the present findings demonstrate that SA-AKI was associated with a decrease in intracellular CSE levels, a concomitant decrease in intracellular H_2_S levels, and exacerbation of ferroptosis. Importantly, the findings indicate the potential benefits of treatment with exogenous H_2_S in terms of alleviation of AKI via inhibition of ferroptosis induced by mitochondrial oxidative stress in renal glomerular endothelial cells during the development of sepsis (Figure 6).

## 4. Materials and Methods

### 4.1. Drugs and Chemicals

GYY4137 (GYY) was obtained from Medchemexpress (HY-107632; Shanghai, China) and dissolved in saline or complete culture medium to the desired concentration 30 min before use. LPS was obtained from Macklin (Shanghai, China). Detection kits for blood urea nitrogen (BUN), creatinine (Cre), and endogenous H_2_S levels were purchased from Jiancheng BioEngineering (Nanjing, China). Detection kits for ROS, malonaldehyde (MDA), GSH, and mitochondrial membrane potential (MMP) were obtained from Solarbio (Beijing, China). The detection kit for Fe^2+^ levels was obtained from Dojindo Molecular Technologies (Shanghai, China). Bicinchoninic acid (BCA) reagent was purchased from Beyotime Biotechnology (Shanghai, China). Ferrostatin-1 (Fer-1) and Mito-TEMPO detection kits (HY-100579 and HY-112879, respectively) were obtained from Medchemexpress (Shanghai, China).

### 4.2. Animals and Treatments

Male wild-type C57BL/6 J mice (8–10 weeks) were purchased from Shanghai Jihui Experimental Animal Breeding Co. (Shanghai, China), housed, and bred under standard conditions (12-h light/dark cycle) under stable temperature (22 ± 2 °C) and humidity (60%) and given free access to food and water. Renal injury is a very common complication of sepsis. Therefore, SA-AKI model mice were established using the previously reported cecal ligation and puncture (CLP) method [42]. Mice were anesthetized by pentobarbital. After midline laparotomy, the cecum was exposed and we chose to ligate the cecum near the distal end of the cecum to about one-third of the bottom of the cecum of the mice, and after ligation, the cecum was punctured with a No. 4 needle, and two fine holes were made with the needle, and a small amount of feces was squeezed out from the punctured holes. Finally, the peritoneum and skin were sutured in sequence. The appropriate amount of saline was given according to the body weight of the mice, and the fluid was rehydrated promptly. The mice were randomly divided into three groups (n = 6 for each group): sham, CLP, and CLP + GYY4137. At 30 min after CLP treatment, the CLP + GYY group was intraperitoneally administered GYY dissolved in saline. The sham and CLP group mice received an equal volume of saline, which was also administered intraperitoneally. Serum samples and kidney tissues of the mice were collected after 24 h. All animal studies were carried out according to the Guide for the Care and Use of Laboratory Animals.

### 4.3. Histological Analysis

The right kidney tissues were fixed with paraformaldehyde (4%) for 48 h, embedded in paraffin, cut into sections that were 3 µm thick, stained with H&E, and evaluated under a light microscope (Nikon Eclipse Ci-L; Nikon, Tokyo, Japan).

### 4.4. Transmission Electron Microscopy Examination of Kidney Tissues

Kidney samples were dissected and immediately fixed in 4% phosphate glutaraldehyde. Each sample was dehydrated, permeabilized, embedded, sectioned to a thickness of 60–80 nm, and mounted. For each sample, five fields of view were randomly selected for imaging under a transmission electron microscope (JEM-1230; Joel, Tokyo, Japan).

### 4.5. Measurement of Serum Endogenous H_2_S, Cre, and BUN Levels

Serum samples were collected from mice of all the groups and stored at −80 °C until they were thawed for the assay. The levels of serum endogenous H_2_S, Cre, and BUN were measured using commercial kits (Jiancheng, Nanjing, China).

### 4.6. Measurement of ROS, GSH, MDA, Fe^2+^, and MMP

The left kidney tissues and serum samples of mice were frozen at −80 °C, and MRGECs were collected for follow-up tests. The expression level of GSH, concentration of MDA, and ROS level were detected with the commercially available kits, Solarbio BC1175, Solarbio BC0020, and Solarbio CA1410, respectively, from Solarbio, China. MMP was detected using the JC-1 assay kit (Solarbio M8650; Beijing, China), and the Fe^2+^ level in the kidney was detected using a commercially available kit (Dojindo Molecular Technologies, Shanghai, China). All indicators were tested by fluorescence spectrophotometer (ND-3300; Thermo Scientific™, Waltham, MA, USA)

### 4.7. Cell Culture and LPS Treatment

The mouse renal glomerular endothelial cell line MRGECs was grown in Dulbecco’s modified Eagle’s medium F12 (Invitrogen, Burlington, ON, Canada) containing 10% fetal calf serum (Bovogen, Australia) at 37 °C in a 95% O_2_/5% CO_2_ atmosphere. MRGECs were randomly divided into three groups: control, LPS, and LPS + GYY4137. LPS was procured from Macklin (Shanghai, China), dissolved in F12 medium at a concentration of 1 mg/mL (as the stock solution), and stored at −80 °C until use. When the MRGECs reached about 60% confluency, they were treated with LPS under standard cell culture conditions for 24 h. At 30 min after LPS treatment, the LPS + GYY4137 group was treated with GYY dissolved in Dulbecco’s modified Eagle’s medium F12 for 24 h. Cells and cell culture supernatants were then collected for subsequent assays.

### 4.8. Cell Viability Assays

The viability of MRGECs cultured in 96-well plates was measured using Cell Counting Kit-8 (CCK-8) (Dojindo Molecular Technologies, Shanghai, China) according to the manufacturer’s instructions. The absorbance of CCK-8 was determined with a microplate reader at 450 nm.

### 4.9. Western Blot Analysis

Frozen left kidney specimens were dispersed mechanically in cold RIPA buffer. Proteins present in the supernatant were extracted and quantified with the BCA assay. After 12.5% and 10% sodium dodecyl sulfate-polyacrylamide gel electrophoresis, the proteins were blotted onto polyvinylidene fluoride membranes (Macklin, Shanghai, China). After blocking in nonfat milk (5%) for 40 min at room temperature, the membranes were incubated at 4 °C for 12–18 h with antibodies against CSE (1:1000; cat. no. ab151769; Abcam), Gpx4 (1:1000; cat. no. ab125066; Abcam), TF (1:1000; cat. no. ab82411; Abcam), FTH-1 (1:1000; cat. no. ab65080; Abcam), gapdh (1:1000; cat. no. ab181602; Abcam), and β-actin (1:1000; cat. no. ab115777; Abcam). The membranes were washed three times with TBS/Tween Buffer (Epizyme Biomedical Technology Co., Shanghai, China) incubated with Goat Anti-Rabbit IgG Secondary Antibody (1:10,000; cat. no. ab7090; Abcam) at room temperature for 2 h, and developed with enhanced chemiluminescence reagents (Epizyme Biomedical Technology Co., Shanghai, China). Band intensities were quantified using the ImageJ (V1.8.0.112) software.

### 4.10. Statistical Analyses

Results of all data analysis are expressed as mean ± standard deviation (SD). Unless otherwise specified, the data were representative of at least three independent experiments. For comparison of more than two groups, one-way analysis of variance (ANOVA), followed by the Student–Newman–Keuls test for multiple comparisons, was performed. Comparisons between two groups were assessed by the *t*-test. A confidence interval of 95% was used for all statistical tests, and *p* < 0.05 was regarded to be statistically significant.

## 5. Conclusions

In this study, we successfully established the models of SA-AKI in in vivo (CLP treatment) and in vitro (LPS treatment). We intervened for the first time with exogenous H_2_S donor GYY4137 and identified that exogenous H_2_S could alleviate SA-AKI by inhibiting ferroptosis induced by excessive mitochondrial oxidative stress. This study provides certain experimental evidence for the potential therapeutic options of exogenous H_2_S in the treatment of sepsis-induced renal injury, which could offer new ideas and theoretical basis for clinical renal protection research and lay the theoretical foundation for the development of related drugs, with potential application prospects. 

## Figures and Tables

**Figure 1 molecules-28-04770-f001:**
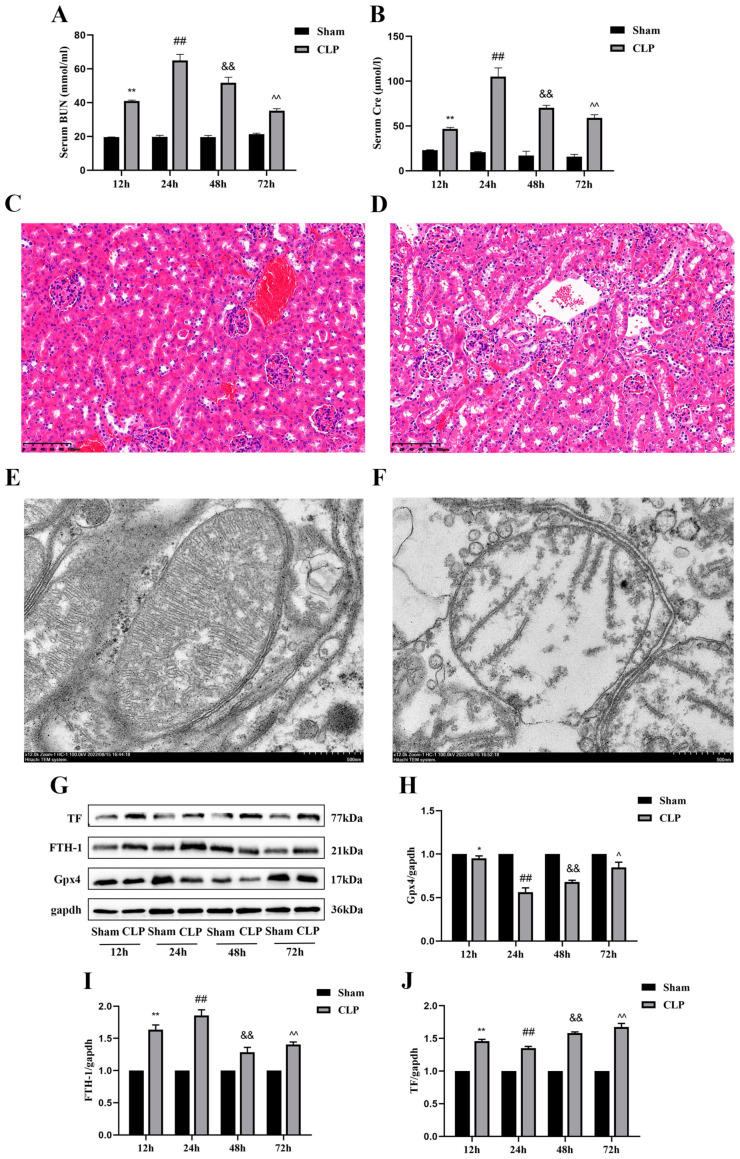
Increase in ferroptosis and AKI in CLP-treated mice compared to the sham group mice. (**A**) BUN levels in the serum at 12, 24, 48, and 72 h in the sham and CLP-treated groups of mice. Data are expressed as mean ± SD (n = 6 per group). ** *p* < 0.01 vs. the sham group at 12 h, ## *p* < 0.01 vs. the sham group at 24 h, && *p* < 0.01 vs. the sham group at 48 h, ^^ *p* < 0.01 vs. the sham group at 72 h. (**B**) Cre levels in the serum at 12, 24, 48, and 72 h in the sham and CLP-treated groups of mice. Data are expressed as mean ± SD (n = 6 per group). ** *p* < 0.01 vs. the sham group at 12 h, ## *p* < 0.01 vs. the sham group at 24 h, && *p* < 0.01 vs. the sham group at 48 h, ^^ *p* < 0.01 vs. the sham group at 72 h. (**C**,**D**) Representative H&E-stained right kidney sections from mice (scale bar = 100 μm) in the sham group (**C**) and CLP group (**D**). (**E**,**F**) Representative TEM image of mitochondria in the right kidney of mice (scale bar = 500 nm) from the sham group (**E**) and CLP group (**F**). (**G**) Representative Western blots of Gpx4, FTH-1, and TF expression in the kidney tissues at 12, 24, 48 and 72 h in the sham and CLP-treated groups of mice. β-actin was used as the internal control. (**H**–**J**) Quantitative analysis of Gpx4 (**H**), FTH-1 (**I**), and TF (**J**) protein expression in the kidney tissues at 12, 24, 48, and 72 h in the sham and CLP-treated groups of mice. Data are expressed as mean ± SD (n = 3 per group). * *p* < 0.05, ** *p* < 0.01 vs. the sham group at 12 h, ## *p* < 0.01 vs. the sham group at 24 h, && *p* < 0.01 vs. the sham group at 48 h, ^ *p* < 0.05, ^^ *p* < 0.01 vs. the sham group at 72 h.

**Figure 2 molecules-28-04770-f002:**
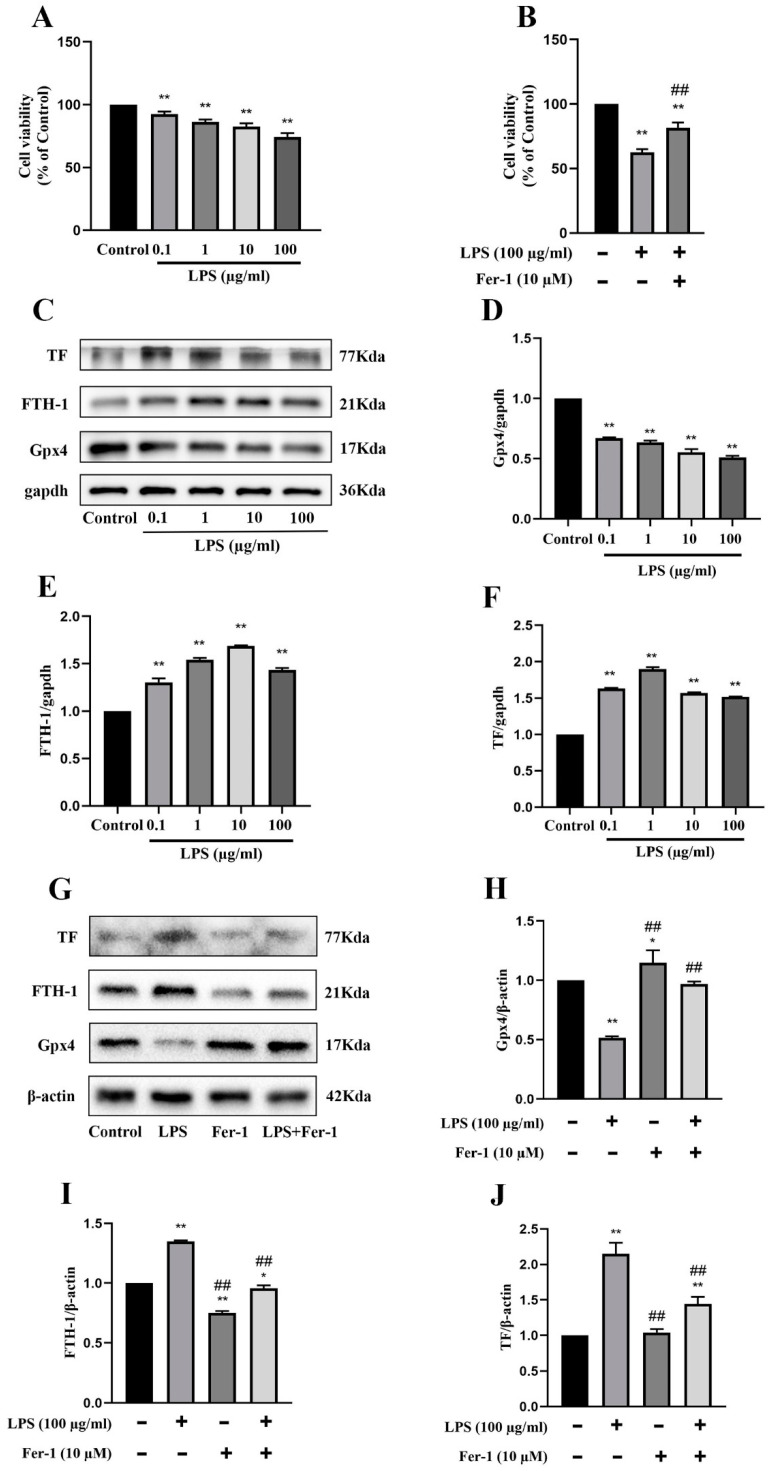
Exacerbation of ferroptosis and AKI in LPS-treated MRGECs. (**A**) MRGECs were incubated with different concentration of LPS (0, 0.1, 1, 10, and 100 µg/mL) for 24 h, and this was followed by CCK-8 detection of cell viability. Data are expressed as mean ± SD (n = 6 per group). ** *p* < 0.01 versus the control group. (**B**) MRGECs were incubated in 100 µg/mL LPS and 10 µM Fer-1 for 24 h, and this was (**C**) followed by detection of cell viability with the CCK-8 assay. Data are expressed as mean ± SD (n = 6 per group). ** *p* < 0.01 versus the control group, ## *p* < 0.01 versus the 100 µg/mL LPS-treated group. Representative Western blots showing Gpx4, FTH-1, and TF expression in the MRGECs under different concentrations of LPS (0, 0.1, 1, 10, and 100 µg/mL). *β*-actin was used as the internal control. (**D**–**F**) Quantitative analysis of Gpx4 (**D**), FTH-1(**E**), and TF (**F**) protein expression in MRGECs under different concentration of LPS (0, 0.1, 1, 10, and 100 µg/mL). Data were expressed as mean ± SD (n = 3 per group). ** *p* < 0.01 versus the control group. (**G**) Representative Western blots for Gpx4, FTH-1, and TF expression in MRGECs treated with 100 µg/mL LPS or 10 µM Fer-1. *β*-actin was used as the internal control. (**H**–**J**) Quantitative analysis of Gpx4 (**H**), FTH-1 (**I**), and TF (**J**) protein expression in MRGECs treated with 100 µg/mL LPS or 10 µM Fer-1. Data are expressed as mean ± SD (n = 3 per group). * *p* < 0.05, ** *p* < 0.01 versus the control group, ## *p* < 0.01 vs. the 100 µg/mL LPS-treated group.

**Figure 3 molecules-28-04770-f003:**
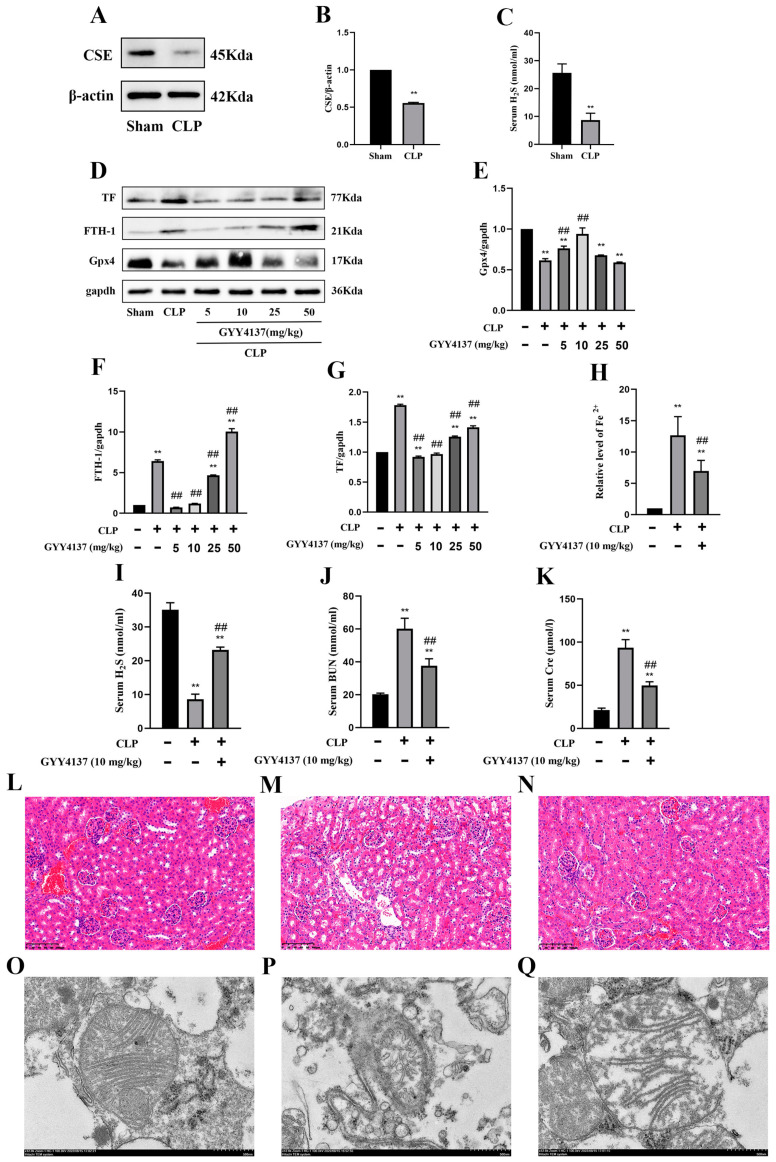
Attenuation of ferroptosis in mice with CLP-induced AKI treated with exogenous H_2_S. (**A**) Representative Western blots for CSE expression in the kidney tissues of the sham and CLP mice. *β*-actin was used as the internal control. (**B**) Quantitative analysis of CSE expression in the kidney tissues of the sham and CLP mice. Data are expressed as mean ± SD (n = 3 per group). ** *p* < 0.01 vs. the sham group. (**C**) H_2_S levels in the serum of the sham and CLP mice. Data are expressed as mean ± SD (n = 6 per group). ** *p* < 0.01 vs. the sham group. (**D**) Representative Western blots for Gpx4, FTH-1, and TF expression in the kidney tissues from mice subjected to CLP and treated with different concentrations of GYY4137 (5, 10, 25, or 50 mg/kg). *β*-actin was used as the internal control. (**E**–**G**) Quantitative analysis of Gpx4 (**E**), FTH-1 (**F**), and TF (**G**) protein expression in the kidney tissues from mice subjected to CLP and treated with different concentrations of GYY4137 (5, 10, 25, and 50 mg/kg). Data are expressed as mean ± SD (n = 3 per group). ** *p* < 0.01 vs. the sham group, ## *p* < 0.01 vs. the CLP group. (**H**) Fe^2+^ levels in the serum of the sham, CLP-treated, and CLP + GYY4137 (10 mg/kg) mice. Data are expressed as mean ± SD (n = 6 per group). ** *p* < 0.01 vs. the sham group, ## *p* < 0.01 vs. the CLP group. (**I**) Serum H_2_S levels in the sham, CLP, and CLP + GYY4137 (10 mg/kg) mice. Data are expressed as mean ± SD (n = 6 per group). ** *p* < 0.01 vs. the sham group, ## *p* < 0.01 vs. the CLP group. (**J**) Serum BUN levels in the sham, CLP, and CLP + GYY4137 (10 mg/kg) groups of mice. Data are expressed as mean ± SD (n = 6 per group). ** *p* < 0.01 vs. the sham group, ## *p* < 0.01 vs. the CLP group. (**K**) Serum Cre levels in the sham, CLP, and CLP + GYY4137 (10 mg/kg) groups of mice. Data are expressed as mean ± SD (n = 6 per group). ** *p* < 0.01 vs. the sham group, ## *p* < 0.01 vs. the CLP group. (**L**–**N**) Representative H&E-stained right kidney sections of mice (scale bar = 100 μm) from the sham group (**L**), CLP group (**M**), and CLP + GYY4137 (10 mg/kg) group (**N**). (**O**–**Q**) Representative TEM images of mitochondria in the right kidney tissue of mice (scale bar = 500 nm) from the sham group (**O**), CLP group (**P**), and CLP + GYY4137 (10 mg/kg) group (**Q**).

**Figure 4 molecules-28-04770-f004:**
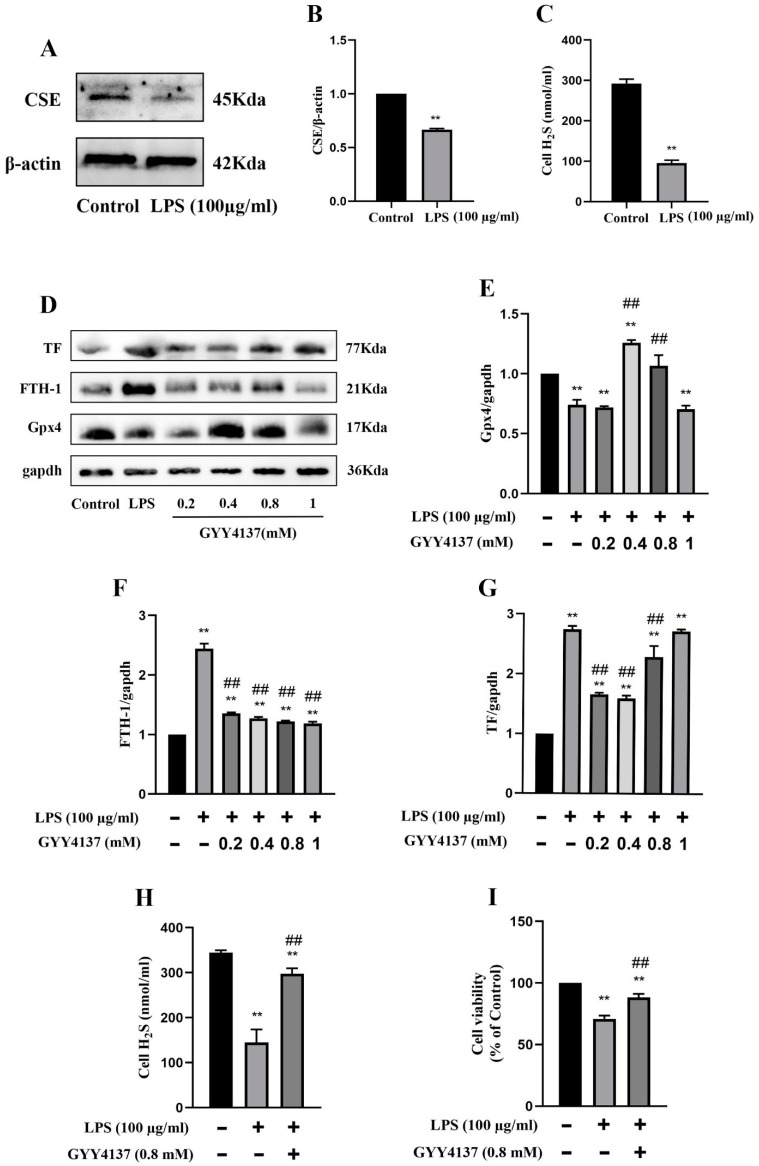
Attenuation of ferroptosis in MRGECs with LPS-induced AKI treated with exogenous H_2_S. (**A**) Representative Western blots for CSE expression in MRGECs of the control and 100 µg/mL LPS-treated groups. *β*-actin was used as the internal control. (**B**) Quantitative analysis of CSE expression in MRGECs of the control and 100 µg/mL LPS-treated groups. Data are expressed as mean ± SD (n = 3 per group). ** *p* < 0.01 vs. the control group. (**C**) H_2_S levels in MRGECs of the control and 100 µg/mL LPS-treated groups. Data are expressed as mean ± SD (n = 6 per group). ** *p* < 0.01 vs. the control group. (**D**) Representative Western blots for Gpx4, FTH-1, and TF expression in MRGECs of the control, 100 µg/mL LPS-treated, and 100 µg/mL LPS + GYY4137 (0.2, 0.4, 0.8, or 1 mM) treatment groups. *β*-actin was used as the internal control. (**E**–**G**) Quantitative analysis of Gpx4 (**E**), FTH-1 (**F**), and TF (**G**) protein expression in the MRGECs. Data are expressed as mean ± SD (n = 3 per group). ** *p* < 0.01 vs. the control group, ## *p* < 0.01 vs. LPS (100 µg/mL). (**H**) H_2_S levels in the MRGECs of the control, 100 µg/mL LPS-treated, and 100 µg/mL LPS + GYY4137 (0.8 mM) treatment groups. Data are expressed as mean ± SD (n = 6 per group). ** *p* < 0.01 vs. the control group, ## *p* < 0.01 vs. the 100 µg/mL LPS-treated group. (**I**) MRGECs were incubated in 100 µg/mL LPS and 0.8 mM GYY4137 for 24 h, and this was followed by detection of cell viability with the CCK-8 assay. Data are expressed as mean ± SD (n = 6 per group). ** *p* < 0.01 vs. the control group, ## *p* < 0.01 vs. the 100 µg/mL LPS-treated group.

**Figure 5 molecules-28-04770-f005:**
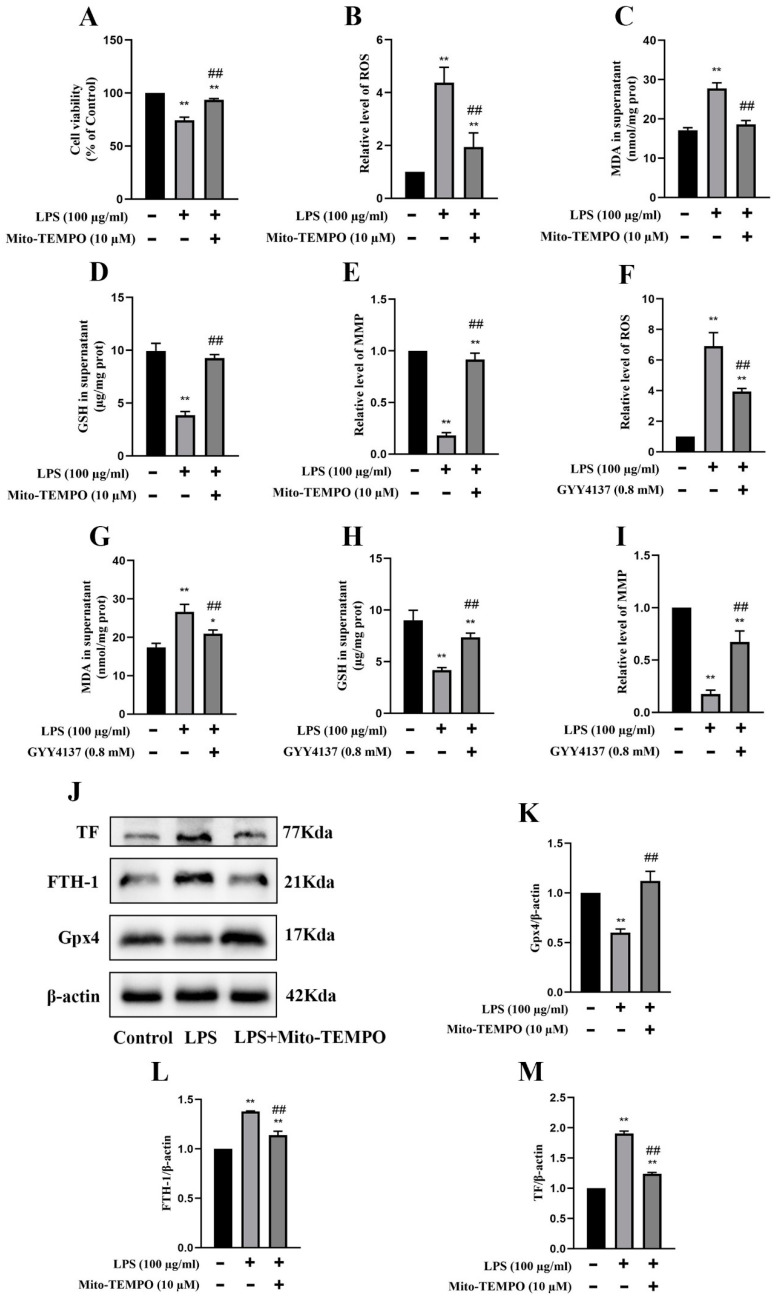
Attenuation of mitochondrial oxidative stress and inhibition of ferroptosis in MRGECs with LPS-induced AKI treated with exogenous H_2_S. (**A**) MRGECs were incubated with 100 µg/mL LPS and 10 µM Mito-TEMPO for 24 h, and this was followed by detection of cell viability by CCK-8. Data are expressed as mean ± SD (n = 6 per group). ** *p* < 0.01 vs. the control group, ## *p* < 0.01 vs. the 100 µg/mL LPS-treated group. (**B**–**E**) Relative level of ROS (**B**), MDA (**C**), GSH (**D**), and MMP (**E**) in the supernatant of MRGECs treated with 100 µg/mL LPS and 10 µM Mito-TEMPO for 24 h. Data are expressed as mean ± SD (n = 6 per group). * *p* < 0.05, ** *p* < 0.01 vs. the control group, ## *p* < 0.01 vs. the 100 µg/mL LPS-treated group. (**F**–**I**) Relative level of ROS (**F**), MDA (**G**), GSH (**H**), and MMP (**I**) in the supernatant of MRGECs treated with 100 µg/mL LPS and 0.8 mM GYY4137 for 24 h. Data are expressed as mean ± SD (n = 6 per group). ** *p* < 0.01 vs. the control group, ## *p* < 0.01 vs. the 100 µg/mL LPS-treated group. (**J**) Representative Western blots for Gpx4, FTH-1, and TF expression in MRGECs of the control, 100 µg/mL LPS-treated, and 100 µg/mL LPS + 10 µM Mito-TEMPO treatment groups. *β*-actin was used as the internal control. (**K**–**M**) Quantitative analysis of Gpx4 (**K**), FTH-1 (**L**), and TF (**M**) protein expression in MRGECs of the control, 100 µg/mL LPS-treated, and 100 µg/mL LPS + 10 µM Mito-TEMPO treatment groups. Data are expressed as mean ± SD (n = 3 per group). ** *p* < 0.01 vs. the control group, ## *p* < 0.01 vs. the 100 µg/mL LPS-treated group.

**Figure 6 molecules-28-04770-f006:**
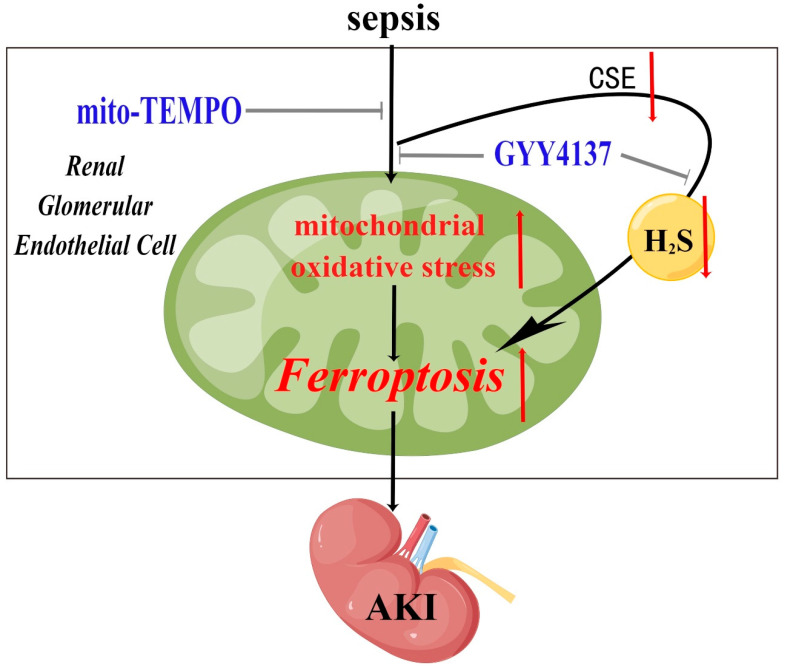
A graphical illustration for the possible mechanisms underlying sepsis resulting in ferroptosis and AKI, and the role of exogenous H_2_S in alleviating SA-AKI by inhibiting ferroptosis triggered by excessive mitochondrial oxidative stress. Sepsis results in the down-regulation of CSE, which results in significantly decreased endogenous H_2_S and excessive mitochondrial oxidative stress. Excessive mitochondrial oxidative stress triggers ferroptosis in renal glomerular endothelial cells, which aggravates AKI. These effects were significantly attenuated by treatment of H_2_S donor GYY4137.

## Data Availability

The original contributions presented in the study are included in the article, further inquiries can be directed to the corresponding author.

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
