# Peer review of "Attenuation of Sepsis-Induced Acute Kidney Injury by Exogenous H_2_S via Inhibition of Ferroptosis"

_molecules, 2023, doi:10.3390/molecules28124770_

Round 1
Reviewer 1 Report
In this study, the authors point out the importance of inhibiting ferroptosis to attenuate sepsis induced kidney injury. This study is well performed using relevant in vitro and in vivo models and will be of interest of researcher in the field. I have nonetheless some major comments to address before endorsing publication:
1) The rationale of using GYY as an H2S exogenous donor rather than others such as NaHS is missing. This raises the question of whether NaHS or other donor mimmic the effect of GYY.
2) The authors claim that “the results showed that 100 µg/ml LPS treatment significantly inhibited MRGECs proliferation compared to treatment with other concentrations of LPS (that is, 0, 0.1, 1, and 10 µg/ml) at 24 h” referring to Figure 2A. However, Figure 2A shows cell viability that is decreased upon treatment most likely due to ferroptosis. The authors have to change the statement in the text. Furthermore, what is the effect of ferroptosis inhibitor on cell viability? Does it rescue the effect of LPS on this particular parameter?
3) The authors only tested ferroptosis as a source of cell death. Does LPS trigger other cell death such as apoptosis or necrosis?
4) The effect of LPS on cell viability looks very limited, at least on Figure 1a. Can the authors modify the scale of the graph to have more numbers between 50 and 100 to have a better idea of the effect of LPS?
5) Can the authors comment on the lack of effect of GYY at high doses in Figure 3?
6) The authors should indicate the number of biological replicates they performed and test the distribution to determine if they can use a parametric test.
Author Response
We thank reviewer #1 for their comments on the article.
My responses to the reviewer's comments and suggestions, one by one, are as follows.
Point 1: The rationale of using GYY as an H2S exogenous donor rather than others such as NaHS is missing. This raises the question of whether NaHS or other donor mimmic the effect of GYY.
Response 1: We thank reviewer #1 for noticing this. The use of GYY4137, a novel exogenous H2S retarder, was selected for this study as a treatment for renal injury in sepsis based on its ability to release H2S slowly and at low concentrations, as well as its long-term stability of release efficacy. Currently, the most common H2S donors in biological studies are sulfide salts, sodium hydrosulfide (NaHS) and sodium sulfide (Na2S), providing direct and immediate biologically relevant forms of sulfides (H2S and HS-), and these salts have been widely used to assess the therapeutic potential of exogenous H2S. However, NaHS rapidly releases large amounts of H2S upon dissolution in water, is extremely unstable, has an unpleasant odor, and has cytotoxic effects. The application of NaHS in the available literature is often inconsistent or difficult to reproduce experiments, probably due to its unstable properties. Therefore, GYY4137 is considered to be an exogenous H2S donor with good water solubility by slow release of H2S through hydrolysis, and it is seen that the effect of H2S release from GYY4137 is more stable and persistent than that of common sulfide salts.
Point 2: The authors claim that “the results showed that 100 µg/ml LPS treatment significantly inhibited MRGECs proliferation compared to treatment with other concentrations of LPS (that is, 0, 0.1, 1, and 10 µg/ml) at 24 h” referring to Figure 2A. However, Figure 2A shows cell viability that is decreased upon treatment most likely due to ferroptosis. The authors have to change the statement in the text. Furthermore, what is the effect of ferroptosis inhibitor on cell viability? Does it rescue the effect of LPS on this particular parameter?
Response 2: We thank reviewer #1 for noticing this. In the Result 2.2 section of the article, we have changed the statement as suggested by the reviewer, and we have added the relevant experiment on the effect of ferroptosis inhibitor Fer-1 on the cell viability of MRGECs stimulated by 100 µg/ml LPS. The results showed that ferroptosis inhibitor could significantly increase the cell viability of 100 µg/ml LPS induced MRGECs in sepsis, and the relevant statements have been supplemented in the Result 2.2 section.
Point 3: The authors only tested ferroptosis as a source of cell death. Does LPS trigger other cell death such as apoptosis or necrosis?
Response 3: We thank reviewer #1 for noticing this. Many studies have shown that LPS-stimulated cells can induce necrosis and apoptosis, the mechanism of which has been widely reported. Ferroptosis is a newly discovered mode of cell death, and it has been reported that ferroptosis may be a key mechanism causing sepsis induced kidney failure. However, the mechanism by which ferroptosis plays a role in septic kidney cells is rarely reported. Renal disease remains by far the leading cause of high morbidity and mortality worldwide, and sepsis induced renal damage will further increase mortality. Therefore, targeted ferroptosis therapy will be a novel treatment for sepsis induced renal disease. Therefore, we constructed an animal model of sepsis in mouse by CLP, and the sepsis cell model was constructed by LPS-induced MRGECs in vitro. Through the detection of ferroptosis markers, it was found that ferroptosis significantly increased in sepsis renal cells, indicating that ferroptosis played an important role in renal damage in sepsis.
Point 4: The effect of LPS on cell viability looks very limited, at least on Figure 1a. Can the authors modify the scale of the graph to have more numbers between 50 and 100 to have a better idea of the effect of LPS?
Response 4: When MRGECs were stimulated at 0.1 µg/ml LPS, the cell viability began to decline, and the viability of MRGECs decreased significantly at 10 µg/ml, while the most significant decrease was observed at 100 µg/ml. In addition, compared with other concentrations, the degree of ferroptosis of MRGECs was also the most serious under LPS stimulation of 100 µg/ml, suggesting that the degree of renal cells damage was the most serious under LPS stimulation of 100 µg/ml. According to the suggestions of the reviewers, more concentrations of stimulation were performed in the LPS concentration range of 50-100 µg/ml, and it was found that the difference was not significant, and the renal cells viability decreased most significantly under the LPS stimulation of 100 µg/ml.
Point 5: Can the authors comment on the lack of effect of GYY at high doses in Figure 3?
Response 5: We thank reviewer #1 for noticing this. It has been reported that appropriate concentration of H2S plays a positive regulatory role in the cell, but high concentration of GYY4137 will release a large amount of H2S in the cell, and high concentration of H2S is cytotoxic. Therefore, when GYY4137 is in high concentration, the effect of reducing renal cells damage in sepsis is weakened, and even has reverse effect.
Point 6: The authors should indicate the number of biological replicates they performed and test the distribution to determine if they can use a parametric test.
Response 6: We thank reviewer #1 for noticing this, and we have indicated and supplemented the reviewer's suggestions and related notes in the section 4.10. Statistical analyses and in all figure legends.

Reviewer 2 Report
Two minor points.
1. Abstrtact. I think that in the sentense "In the vitro experiments, LPS..." the authors lost one "in" (In the in vitro experiments...)
2. Materials and Methods. I think that despite the existance of the reference, a brief description of the surgical procedure of cecal ligation and punction is necessary.
Author Response
We thank reviewer #2 for their comments on the article.
My responses to the reviewer's comments and suggestions, one by one, are as follows.
Point 1: Abstrtact. I think that in the sentense "In the vitro experiments, LPS..." the authors lost one "in" (In the in vitro experiments...)
Response 1: We thank reviewer #2 for noticing this, and we have corrected this problem in the Abstract.
Point 2: Materials and Methods. I think that despite the existance of the reference, a brief description of the surgical procedure of cecal ligation and punction is necessary.
Response 2: We thank reviewer #2 for noticing this, and we have added a surgical procedure of cecal ligation and puncture in the 4.2 Animals and treatments section of the acticle.

Round 2
Reviewer 1 Report
The authros properly addressed my comments.